# Non-Local Seismo-Dynamics: A Fractional Approach

**Vladimir Uchaikin** *,†  **and Elena Kozhemiakina** †

Department of Theoretical Physics, Ulyanovsk State University, Lev Tolstoy Street, 42, 432017 Ulyanovsk, Russia
* Correspondence: vuchaikin@gmail.com
† These authors contributed equally to this work.

**Abstract:** This paper consists of a general consideration of a seismic system as a subsystem of another, larger system, exchanging with it by extensive dynamical quantities in a sequential push mode. It is shown that, unlike an isolated closed system described by the Liouville differential equation of the first order in time, it is described by a fractional differential equation of a distributed equation in the interval (0, 1] order. The key characteristic of its motion is a spectral function, representing the order distribution over the interval. As a specific case of the process, a system with single-point spectrum is investigated. It follows the fractional Poisson process method evolution, obeying via a time-fractional differential equation with a unique order. The article ends with description of statistical estimation of parameters of seismic shocks imitated by Monte Carlo simulated fractional Poisson process.

**Keywords:** aftershocks; power laws; fractional equations; open systems; distributed orders; fractional Poisson process; Levy statistics; relaxation spectra

## 1. Introduction

Dynamic processes in the Earth's crust, studied by seismology, are undoubtedly a random process; however, the empirical laws of Gutenberg–Richter and Omori–Utsu, governing the spatio-temporal distribution of aftershocks, are expressed through power functions, demonstrating special and unusual properties for classical statistical systems such as heredity (memory), when considering a time variable, and fractality (self-similar inhomogeneity), if the spatial distribution of events is considered. Being the basis of the popular ETAS (epidemic-time aftershocks) model [1], they are confirmed both in systematic observations and in specific mechanical models, but we aim to seek links between the scale-invariant properties and more fundamental concepts, such as the critical self-organization [2].

As in the theory of electrical circuits, the relaxation mechanisms of complex mechanical systems (including continuum ones) are represented by drawing up diagrams consisting of individual elements (blocks). The use of serial, parallel or combined orders of their connections makes it possible to describe rheological processes in systems of varying complexity. The mathematical design of such a process is conveniently carried out in terms of Fourier transforms, with each element being assigned its own Fourier image. This approach proved fruitful not only in materials science and structural rheology, but also in seismodynamics (see [3]).

Figure 1 shows a scheme of such kind, including slider-blocks, mimicking a non-Debye viscoelastic lower crust and upper mantle interaction, exhibiting elements of power-law rheology.

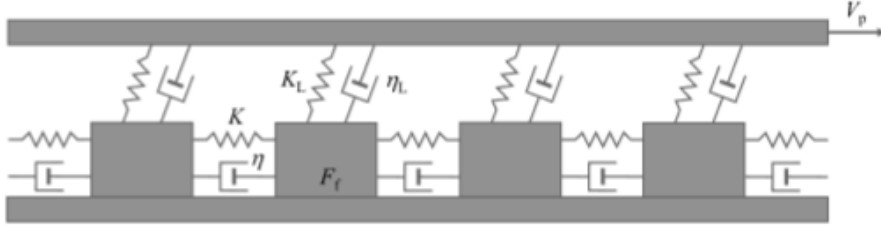

**Figure 1.** One-dimensional sketch of the sider-blocks model [4].

Such a way of reasoning by composing a system scheme of a few elements (like in electrical circuit engineering) is widespread in complex system rheology. Viscoelastic elements take place (in its behaviour) between Hook's elasticity and Newtonian viscosity (springpots). An example of such a scheme is the ladder model (Figure 2). The figure clearly shows the idea of self-similarity (fractality). The new element (springpot ) was introduced by Koeller [5] as a phenomenological description of viscoelastic behaviour, in terms of a complex modulus of the form

$$E^* = E(i\omega\tau)^\beta = K(i\omega)^\beta,$$

where $K = E\tau^\beta$. This $\beta$-element) describes behaviour intermediate between linear elasticity ($\beta = 0$) and viscosity ($\beta = 1$). The authors of [4] remark that "just as pure viscous flow does not define any particular timescale, the element describes self-similar behaviour on all timescales" (see, for details [6]).

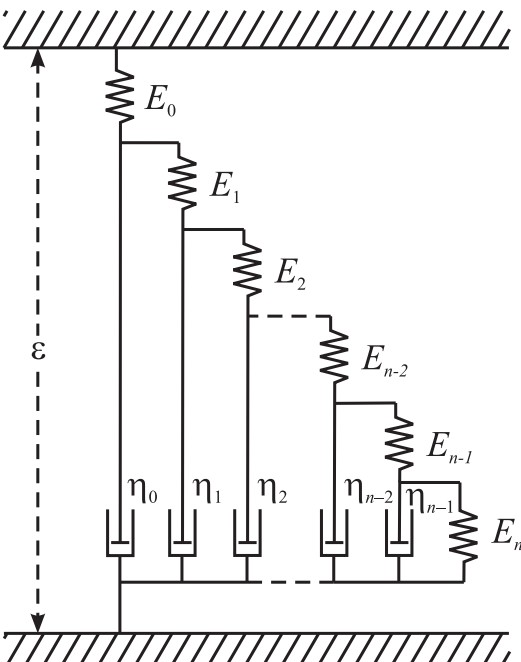

**Figure 2.** Ladder model with scaling behaviour, leading to scaling laws for the complex (shear) modulus, usually with a power law exponent $\alpha = 1/2$ [7].

These general information is obviously not enough to predict such events. From the geophysical point of view, many aspects of these processes are in principle understandable and even sufficiently studied. However, the classical mechanical approach requires additional information about the environment and initial conditions. The main problem, it seems to us, remains in the development of a general approach for this accounting. Due to the natural diversity of seismically hazardous areas, one can talk about considering a statistical ensemble of similar situations, that is, about applying the methods of statistical

mechanics. One of these is the mechanics of open systems, developed on the basis of Zwanzig–Mori projection operators formalism applying to the Liouville equation, which describes the behavior of open systems in contact with a nonequilibrium environment. The results presented in this paper indicate that this way leads to interesting theoretical conclusions about the fractional nature of the dynamic equations of such a system, including information about both its seismological prehistory (memory phenomenon) and the multiscale heterogeneity (fractal phenomenon) of the environment. As a result, the usual (Gaussian) statistical paradigm is replaced by the concept of stable limit distributions with asymptotic properties consistent with the basic empirical laws of seismodynamics. A typical example is the fractal generalization of Poisson's law presented in the paper, which is in qualitative agreement with the observed dynamics of aftershocks. Very important for understanding seismicity is the famous Burridge–Knopov model (a kind of springblock model, Figure 1). This model is about self-oscillations in nonlinear dissipative dynamicsystems which are inherently closer to the nature of seismicity than Hamiltonian systems, and afractional approach can also be applied to them [8]. Moreover, with regard to originality, our approach resonates with the model developed in paper [9], which also deals with a fractional model, constructing an Empirical Cumulative Foreshock Waiting Time Distribution Function. The main idea of our article is to link the agreement of such results with empirical laws by using a general statistics–mechanical approach. Doing this, we involve fundamental concepts such as complexity, self-similarity and Levy statistics.

## 2. Complexity, Self-Similarity and Levy-Statistics

### 2.1. Complexity

Several of the above examples demonstrate an important fact for understanding the essence of the matter: averaging exponential distributions over a parameter generates power-type distributions. Recall that the exponential distribution has the property of no aftereffect: the conditional distribution coincides with the unconditional one up to a normalization factor. Thus, the probability of a cosmic ray proton colliding with a nucleus in the interval $(t, t + dt)$ (in the Fermi model) does not depend on $t$. This property is often formulated as a lack of memory and is considered quite natural for most of the phenomena of the surrounding world. Whenever we say "increment $I$ proportional to $Idt$", we use this property by default. Additionally, now, as a result of averaging over a parameter, the exponentially distributed value loses this property, becoming, as Volterra said, hereditary, possessing the memory. First of all, this affects the need to take into account the prehistory of the system up to the time $t$, if we are interested in further events. In non-hereditary mechanics, the further evolution of the system is determined by the state at one initial moment of time; in order to predict the evolution of a hereditary system, one must know its entire prehistory. Power laws just characterize the behavior of such systems.

Concerning the complexity problem, B. West writes in his Colloquium [10] "Specifically, the complexity of nonlinear dynamic phenomena demands that we extend our horizons beyond analytic functions and analysis suggests that the functions of interest lack traditional equations of motion. To explore this lack we introduce fractional thinking, which is a kind of in-between thinking; between the integer-order moments, such as the mean and variance, there are fractional moments required when empirical integer moments fail to converge; between the integer dimensions there are the fractal dimensions that are important when data have no characteristic scale length; and between the integer-valued operators that are local in space and time, are the noninteger operators necessary to describe dynamics that have long-time memory and spatial heterogeneity. Complex phenomena require new ways of thinking and the fractional calculus provides one framework for that thinking [5]".

Speaking about the mechanics or thermodynamics of disordered media, it is impossible to get around the problem of adding random variables, which arises every time. When it comes to additive functionals of a system, such as kinetic energy, entropy, mass, the central limit theorem, the main tool in this mathematical procedure, assumes the mutual

independence of random terms, which greatly limits the scope of its applicability. The second limitation is related to the limited variance of terms. Thus, the total energy of photons in a beam with a Lorentzian frequency spectrum $f(\omega) = \frac{A}{B^2+(\omega-\omega_0)^2}$ should not be approximated by a Gaussian density, since the dispersion of an individual term is infinite. Those distributions, the dispersion of which diverges, turn out to be outside the scope of the CLT, and it is difficult to imagine how the physics of disordered structures (including geophysics) would develop further if the outstanding French mathematician *P*. Levy had not discovered that this theorem can not only be preserved, but also significantly expands the field of its application, replacing the requirement of boundedness of the variance with the property of self-similarity of distributions, their self-similarity with respect to linear transformations of arguments.

*2.2. Self-Similarity*

Self-similarity (scale invariance, scaling) is a particular type of symmetry that makes it possible to compensate a rescaling of some variables by a corresponding rescaling of others [11]. The self-similarity of a time-dependent distribution density $p(x,t)$ can be expressed as follows:

$$p(x,t) = t^{-H}p(xt^{-H},1).$$

In terms of the random variable, this property assumes the form

$$X(t) \stackrel{d}{=} t^H X(1),$$

or

$$X(at) \stackrel{d}{=} a^H X(t),$$

for $a > 0$ and any fixed $t$. Generally, a random process $\{X(t)\}$ is called self-similar if the distributions of all its finite-dimensional vectors are self-similar:

$$\{X(at_k)\}_{k=1}^n \stackrel{d}{=} \{a^H X(t_k)\}_{k=1}^n.$$

For the self-similarity of a homogeneous Markovian process, however, the self-similarity of the one-dimensional distribution is sufficient.

We now define an *L*-process as a homogeneous process with a self-similar one-dimensional distribution

$$p(x,t) = t^{-1/\alpha}g^{(\alpha)}(xt^{-1/\alpha}),$$

where $\alpha = 1/H$, and $g^{(\alpha)}(x)$ is the density of the distribution, not yet known. Thus, we can pass from a *W* process to an *L* process by replacing the finite-variance condition with the scaling requirement.

Obviously, such a replacement cannot narrow the class of processes considered, since a W process satisfies this condition and thus remains in the *L* class with a $\alpha = 2$. Can any new processes appear in this case? The answer depends on the existence of non-Gaussian *L* processes satisfying the scaling condition.

Let us consider two instants of time, $t$ and $t + \tau$. The random coordinates of the L process at these times are related as follows:

$$X(t + \tau) = X(t) + X(\tau).$$

If $X(0) = 0$, the random variables $X(t)$ and $X(\tau)$ are the increments of the process over the nonintersecting intervals $(0,t)$ and $(t,t+\tau)$, thus being independent. The distribution density for their sum is the convolution of the densities for the two summands:

$$p(x,t+\tau) \equiv p(x,t) * p(x,\tau) \equiv \int_{-\infty}^{\infty} p(x-x',t)p(x',\tau)dx'.$$

It is convenient to pass from the densities to the characteristic functions

$$\tilde{p}(k,t) = \langle \exp\{ikX(t)\}\rangle = \int_{-\infty}^{\infty} e^{ikx} p(x,t)dx,$$

for which the convolution reduces to the multiplication

$$\tilde{p}(k,t+\tau) = \tilde{p}(k,t)\tilde{p}(k,\tau),$$

and the scaling condition assumes the form

$$\tilde{p}(k,t) = \tilde{g}^{(\alpha)}\left(kt^{1/\alpha}\right).$$

Combining the two formulas, we obtain

$$\tilde{g}^{(\alpha,\beta)}(k) = \exp\{-|k|^{\alpha}[1 - i\beta\tan(\alpha\pi/2)\operatorname{sign}k]\}, \quad -\infty < k < \infty. \tag{1}$$

This is a standard representation of the stable characteristic function in form (1). The so-called form (2) is often used with a different choice of the skew parameter

$$\tilde{g}(k;\alpha,\theta) = \exp\{-|k|^{\alpha}\exp\{-i(\theta\alpha\pi/2)\operatorname{sign}k\}\}. \tag{2}$$

Observe that case with $\beta = 0$ relates to the symmetric distriburion (normal, if $\alpha = 2$), and case $\beta = 1$ corresponds to one-sided distribution on posiitive semiaxes, denoted by $g_+(x;\alpha)$ and concentrated at $x = 1$, if $\alpha = 1$ (see Figure 3).

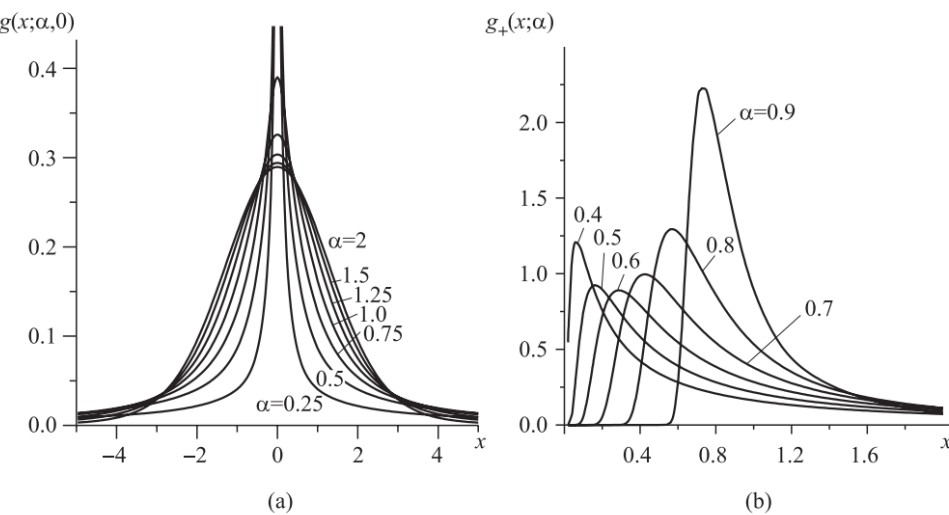

**Figure 3.** Symmetric (**a**) and one-sided (**b**) one-dimensional Levy-stable densities.

The most important circumstances are the facts that Levy-stable distributions are limiting in the class of distributions with power tails, and that their densities are solutions of differential equations of fractional orders [12,13].

## 3. Open Subsystem Dynamics

### 3.1. The Statistical Approach to the Problem

There exist many physical situations in classical equilibrium thermodynamics where one of the two subsystems, say, subsystem 1, is assumed to be large enough so that the influence of this exchange (which occurs through the surface and is therefore proportional to the surface/volume ratio) can be considered small. At the same time, only a small fraction of these already small flows contains information previously taken out of subsystem 1 and now returned to it, so that in the end we are dealing with a small value of the second

order, which justifies the usual neglect of hereditarity in macroscopic (albeit far from all) processes. When moving to small (meso- and nano-) sizes, the surface volume ratio changes and it may be necessary to take into account hereditary effects [14]. Thus, there are two ways to predict the evolution of an open system that is part of a closed one. The first way is by solving the differential equation for the evolution of a closed system under given initial conditions, extracting from the resulting solution all the information related to this subsystem. The second way is to solve the integro-differential equation for the most open subsystem, without involving its environment in the solution process. The latter does not mean that we ignore the environment: the integral term just describes the transfer of information from an open system in early times through its environment to it in later times. Here, "weave" information about the initial state of the environment is absent.

Assume now that the whole system is inaccessible to observation, it is possible to see only a certain part of it, which, however, exchanges with the inaccessible part in the course of its evolution, imparting to it energy and other dynamic characteristics of an extensive type. The simplest example of this kind is the standard thermodynamic problem: one subsystem is immersed in another and both have the possibility of energy, mass, entropy. The object of observation is the smaller of them, and it is assumed that its changes do not affect the behavior of the external subsystem surrounding it, called the thermostat. Under laboratory conditions, it is very easy to carry out such an experiment. However, in the case of an earthquake, we cannot even mentally separate these subsystems, so we have to resort to very simplified schemes such as the one described in Section 1. Here, we will try to consider this problem, remaining within the framework of thermodynamic phenomenology, without neglecting, however, the reverse influence of the subsystem under study on the environment, more precisely, representing this influence as a sequence of impulses (impacts).

Consider a closed Hamiltonian system characterized by a set of phase variables $\{q_1, p_1; \ldots; q_n, p_n\}$. Let us break it into two parts: subsystem 1 with coordinates $\mathbf{x} \equiv \{q_1, p_1; \text{о}ts; q_{n_1}, p_{n_1}\}$ and subsystem 2 with the remaining coordinates, the set of which we denote by $\mathbf{y}$. The Hamiltonian of the original system will appear as the sum

$$\mathcal{H}(\mathbf{x}, \mathbf{y}) = \mathcal{H}_1(\mathbf{x}) + \mathcal{H}_2(\mathbf{y}) + \mathcal{H}_{12}(\mathbf{x}, \mathbf{y}), \tag{3}$$

the first term in which represents the Hamiltonian of subsystem 1 in the absence of subsystem 2, the second is the Hamiltonian of subsystem 2 in the absence of subsystem 1, and the third is its interaction Hamiltonian. If a $(\mathbf{x}_0, \mathbf{y}_0)$ is a deterministic initial state of the system, and $\mathbf{X}(t; \mathbf{x}_0, \mathbf{y}_0)$ and $\mathbf{Y}(t; \mathbf{x}_0, \mathbf{y}_0)$ are the phase trajectories of subsystems, parameterized with common time $t$, then the phase probability density

$$f_{n_1+n_2}(q_1, p_1; \ldots; q_{n_1+n_2}, p_{n_1+n_2}; t) \equiv f(\mathbf{x}, \mathbf{y}, t)$$

reads as

$$f(\mathbf{x}, \mathbf{y}, t) = \delta[\mathbf{x} - \mathbf{X}(t; \mathbf{x}_0, \mathbf{y}_0)]\delta[\mathbf{y} - \mathbf{Y}(t; \mathbf{x}_0, \mathbf{y}_0)].$$

It is more naturally, however, to imagine a statistical ensemble of initial states $\{\mathbf{X}_0, \mathbf{Y}_0\}$, characterized by the density

$$f(\mathbf{x}, \mathbf{y}, t_0) = \langle \delta(\mathbf{x} - \mathbf{X}_0)\delta(\mathbf{y} - \mathbf{Y}_0) \rangle. \tag{4}$$

This function obeys the Liouville equation, we will write which as

$$\frac{\partial f}{\partial t} = \mathsf{L} f(\mathbf{x}, \mathbf{y}, t) \tag{5}$$

with the initial condition (4). In conformity with Equation (3), the Liouvillian given by formula

$$\mathsf{L}f = -\{\mathcal{H}, f\} \equiv -\sum_{i=1}^{n} \left( \frac{\partial \mathcal{H}}{\partial p_i} \frac{\partial f_n}{\partial q_i} - \frac{\partial \mathcal{H}}{\partial q_i} \frac{\partial f_n}{\partial p_i} \right),$$

into three terms

$$\mathsf{L} = \mathcal{L}_1 + \mathcal{L}_2 + \mathcal{L}_{12}, \tag{6}$$

containing, respectively, $\mathcal{H}_1$, $\mathcal{H}_2$ and $\mathcal{H}_{12}$. In the absence of interaction between subsystems, the operator $\mathcal{H}_{12}$ vanishes, and correlations between subsystems are determined only by the initial state. If there are none,

$$f(\mathbf{x}, \mathbf{y}, t_0) = f_1(\mathbf{x}, t_0) f_2(\mathbf{y}, t_0),$$

then the solution of Equation (5), marked for this case by superscript 0, has the form of a product of the functions

$$f^0(\mathbf{x}, \mathbf{y}, t) = f_1(\mathbf{x}, t) f_2(\mathbf{y}, t), \tag{7}$$

each of which obeys its own equation,

$$\frac{\partial f_1}{\partial t} = \mathcal{L}_1 f_1(\mathbf{x}, t),$$

$$\frac{\partial f_2}{\partial t} = \mathcal{L}_2 f_2(\mathbf{y}, t),$$

with its initial condition:

$$f_1(\mathbf{x}, t_0) = \langle \delta(\mathbf{x} - \mathbf{X}_0) \rangle,$$

$$f_2(\mathbf{y}, t_0) = \langle \delta(\mathbf{y} - \mathbf{Y}_0) \rangle.$$

Note, that

$$\int f_1(\mathbf{x}, t) d\mathbf{x} = \int f_2(\mathbf{y}, t) d\mathbf{y} = \int \int f(\mathbf{x}, \mathbf{y}, t) d\mathbf{x} d\mathbf{y} = 1$$

for all $t \geq t_0$.

### 3.2. Zwanzig–Mori Projection Operators

In the presence of interaction between the subsystems, the procedure for transforming into a system of two equations is performed using the Zwanzig–Mori projection operator P. Define the action of the operator P on function $f(\mathbf{x}, \mathbf{y}, t)$ by formula

$$\mathsf{P}f(\mathbf{x}, \mathbf{y}, t) = \int d\mathbf{y}' f(\mathbf{x}, \mathbf{y}', t) \cdot f_2(\mathbf{y}, t_0) = f_1(\mathbf{x}, t) f_2(\mathbf{y}, t_0) \equiv \phi_1(\mathbf{x}, \mathbf{y}, t) \tag{8}$$

and act on both sides of the equality by operator P once more:

$$\mathsf{P}^2 f(\mathbf{x}, \mathbf{y}, t) = \mathsf{P}\left( \int d\mathbf{y}' f(\mathbf{x}, \mathbf{y}', t) \cdot f_2(\mathbf{y}, t_0) \right) =$$

$$= \int d\mathbf{y}' \left( \int d\mathbf{y}'' f(\mathbf{x}, \mathbf{y}'', t) f_2(\mathbf{y}', t_0) \right) \cdot f_2(\mathbf{y}, t_0) =$$

$$= \int d\mathbf{y}'' f(\mathbf{x}, \mathbf{y}'', t) \left( \int f_2(\mathbf{y}', t_0) d\mathbf{y}' \right) \cdot f_2(\mathbf{y}, t_0) =$$

$$= \int d\mathbf{y}'' f(\mathbf{x}, \mathbf{y}'', t) \cdot f_2(\mathbf{y}, t_0) = \phi_1(\mathbf{x}, \mathbf{y}, t).$$

Observe that repeatedly applying the operator did not change the result. For this reason, the operator P is called the projection operator. Define also the complementary operator $P'$ via

$$P' = \mathbf{1} - P \tag{9}$$

(here, 1 stands for identical operator). It is easy to see that $P'$ is also the projection operator, and is orthogonal to P:

$$PP' = P'P = 0 \tag{10}$$

Moreover, it can be shown that P and $P'$ commutate with $\mathcal{L}_1$ and $\mathcal{L}_2$,

$$P\mathcal{L}_1 - \mathcal{L}_1 P = P'\mathcal{L}_1 - \mathcal{L}_1 P' = 0, \quad P\mathcal{L}_2 - \mathcal{L}_2 P = P'\mathcal{L}_2 - \mathcal{L}_2 P' = 0, \tag{11}$$

operators P and $\mathcal{L}_2$ are mutually orthogonal,

$$P\mathcal{L}_2 = \mathcal{L}_2 P = 0, \tag{12}$$

and the equality

$$P\mathcal{L}_{12}P = 0 \tag{13}$$

is valid [14].

### 3.3. Splitting the Liouville Equation

According to (8), the phase space of the total system can be represented as the sum of two terms

$$f(\mathbf{x}, \mathbf{y}, t) = Pf(\mathbf{x}, \mathbf{y}, t) + P'f(\mathbf{x}, \mathbf{y}, t) \equiv \phi_1(\mathbf{x}, \mathbf{y}, t) + \phi_2(\mathbf{x}, \mathbf{y}, t), \tag{14}$$

having following from Equation (9) properties:

$$P\phi_1 = \phi_1, \quad P\phi_2 = 0, \quad P'\phi_1 = 0, \quad P'\phi_2 = \phi_2. \tag{15}$$

Inserting decompositions (6) and (14) into the Liouville Equation (5),

$$\frac{\partial}{\partial t}(\phi_1 + \phi_2) = (\mathcal{L}_1 + \mathcal{L}_2 + \mathcal{L}_{12})(\phi_1 + \phi_2), \tag{16}$$

open the brackets and apply operator P to both sides of the equality:

$$\frac{\partial}{\partial t}P\phi_1 + \frac{\partial}{\partial t}P\phi_2 = P\mathcal{L}_1\phi_1 + P\mathcal{L}_1\phi_2 + P\mathcal{L}_2\phi_1 + P\mathcal{L}_2\phi_2 + P\mathcal{L}_{12}\phi_1 + P\mathcal{L}_{12}\phi_2.$$

Accounting for relations (11)–(13) and (15) allows us to significantly reduce this equation without any losses:

$$\frac{\partial \phi_1}{\partial t} = \mathcal{L}_1\phi_1 + P\mathcal{L}_{12}\phi_2. \tag{17}$$

Let us now return to Equation (16) and act on both its sides by the complement operator $P'$:

$$\frac{\partial}{\partial t}P'\phi_1 + \frac{\partial}{\partial t}P'\phi_2 = P'\mathcal{L}_1\phi_1 + P'\mathcal{L}_2\phi_1 + P'\mathcal{L}_{12}\phi_1 + P'(\mathcal{L}_1 + \mathcal{L}_2 + \mathcal{L}_{12})\phi_2.$$

The first term in left side vanishes because of the third property in list (15), first and second terms in right side change their form with the use of relations (10) and (11):

$$P'\mathcal{L}_1\phi_1 = P'\mathcal{L}_1 Pf = P'P\mathcal{L}_1 f = 0,$$

$$P'\mathcal{L}_2\phi_1 = P'\mathcal{L}_2 Pf = P'P\mathcal{L}_2 f = 0.$$

Taking into account, that

$$P'\frac{\partial \phi_2}{\partial t} = \frac{\partial (P'\phi_2)}{\partial t} \equiv \frac{\partial \phi_2}{\partial t},$$

and returning to the total Liouvillian symbol (6), rewrite the second equation in the form (for the sake of simplicity, we ignore the time-dependence of the effective Liouvillian of subsystem 2; it can be accounted for by using chronological Dyson operator):

$$\frac{\partial \phi_2}{\partial t} = P'L\phi_2 + P'\mathcal{L}_{12}\phi_1.$$

*3.4. Replica Lindblad Equation*

Recalling that the initial condition is assumed to be given at $t_0$, we write the formal solution of the second equation of the system in the form

$$\phi_2(\mathbf{x}, \mathbf{y}, t) = \int\limits_{t_0}^{t} d\tau \exp[(t - \tau)P'L]\mathcal{L}_{12}\phi_1(\mathbf{x}, \mathbf{y}, \tau) + \exp[(t - t_0)P'L]\phi_2(\mathbf{x}, \mathbf{y}, t_0). \quad (18)$$

By property (13), the factor $P'\mathcal{L}_{12}\phi_1$ that appeared under the integral is slightly simplified:

$$P'\mathcal{L}_{12}\phi_1 = (1 - P)\mathcal{L}_{12}Pf = \mathcal{L}_{12}\phi_1.$$

Let us substitute the solution (18) of the second equation into the right side of the first Equation (17) of the system:

$$\frac{\partial}{\partial t}\phi_1(\mathbf{x}, \mathbf{y}, t) = PL\phi_1(\mathbf{x}, \mathbf{y}, t) + PL\int\limits_{t_0}^{t} d\tau \exp[(t - \tau)P'L]\mathcal{L}_{12}\phi_1(\mathbf{x}, \mathbf{y}, \tau) +$$

$$PL\exp[(t - t_0)P'L]\phi_1(\mathbf{x}, \mathbf{y}, t_0).$$

Passing, according to Equation (8), to the phase density of subsystem 1, we obtain the second equation in the form

$$\frac{\partial}{\partial t}f_1(\mathbf{x}, t) = L_1 f_1(\mathbf{x}, t) + \int\limits_{t_0}^{t} d\tau Q_1(t - \tau)f_1(\mathbf{x}, \tau) + M_1(\mathbf{x}, \mathbf{y}, t_0).$$

where

$$L_1 f_1(\mathbf{x}, t) = [f_2(\mathbf{y}, t_0)]^{-1} PL f_2(\mathbf{y}, t_0) f_1(\mathbf{x}, t),$$

$$Q_1(t - \tau)f_1(\mathbf{x}, \tau) = [f_2(\mathbf{y}, t_0)]^{-1} P\mathcal{L} \exp[(t - \tau)P'L]\mathcal{L}_{12} f_2(\mathbf{y}, t_0) f_1(\mathbf{x}, \tau)$$

and

$$M_1(\mathbf{x}, \mathbf{y}, t_0) = [f_2(\mathbf{y}, t_0)]^{-1} PL \exp[(t - t_0)\mathcal{L}]P' f_2(\mathbf{y}, t_0) f_1(\mathbf{y}, t_0).$$

The last term reflects the influence of the initial conditions of the cumulative system on the subsequent motion of subsystem 1. At the moment $t = t_0$, the integral term, which reflects the influence of the prehistory of the process on $\partial f_1/\partial t$, disappears. This simply means that the system did not exist before this moment, in other words, at the moment $t_0$ the system was born. However, such processes are excluded in classical mechanics, so it is logical to take $t_0 = -\infty$ and put

$$\lim_{t_0 \to -\infty} M_1(\mathbf{x}, \mathbf{y}, t_0) = 0.$$

Making a similar substitution in the lower limit of the memory integral, we obtain some phenomenological analog (replica) of the quantum *Lindblad equation* (Replica Lindblad Equation)

$$\frac{\partial}{\partial t} f_1(\mathbf{x}, t) = \mathsf{L}_1 f_1(\mathbf{x}, t) + \int\limits_{-\infty}^{t} d\tau \mathsf{Q}_1(t - \tau) f_1(\mathbf{x}, \tau), \tag{19}$$

behind which we leave name Replica Lindblad Equation (RLE).

Despite the absence of explicit expressions for RLE-operator, the physical meaning of the equation is quite transparent. The change in the state of the observed closed system $1 + 2$ in time-interval $(t, t + dt)$ is completely determined by its state at the moment $t$; therefore, to predict its behavior in the future $(t > t_0)$, it is sufficient to know the state at some any one moment of time $t_0$, the prehistory $(t \in (-\infty < t < t_0))$ does not matter. If only a part of this system is available for observation (the open system 1), then the state of subsystem 1 alone is not enough to predict its movement. The subsystem 1 is (and was earlier) in interaction with subsystem 2, characterized by hidden variables. At some time-intervals, it transferred to subsystem dynamical quantities (momentum, angular momentum, energy), which participated in the evolution of subsystem 2 and after some time came back to subsystem 1. It is this exchange of dynamic characteristics that is reflected by the hereditary integral. In other words, information about the prehistory of an open system is stored in its environment for some time and then returned to itself. This property revealed in the process often termed by word *memory*, although it should not be directly associated with its biological counterpart, which implies the storage of information in the biological object itself.

If we are talking about spatially separated subsystems, and not about those combined in space, such as, for example, electronic and ionic components in a crystal, this dynamic information is transmitted through the surface of the $S$ subsystem. The result depends on the surface/volume ratio. For macroscopic samples, this ratio is small and such an exchange can be neglected (microcanonical ensemble representing the open $S$ subsystem as closed) or limited to the exchange of uncorrelated small portions (canonical and grand canonical ensembles). In both these cases, the integral vanishes, and we obtain the traditional mechanical basis of thermodynamics. With a decrease in the size of the sample, we enter the area of meso- and, further, nanomechanics. The number of "actors" is drastically reduced here (from $10^{23}$ atoms to hundreds of thousands or even just hundreds of atoms). In this case, the role of surface effects in many respects becomes decisive, and the integral associated with them in Equation (19) turns into an equal partner among the other members of the equation.

*3.5. Fractional Version of RLE*

Let us return to RLE—the main equation of the dynamics of open systems (in our approximation)—omitting, for brevity, the indices 1 for all terms and the phase variable $\mathbf{x}$ in the density arguments:

$$\frac{\partial f}{\partial t} = \mathsf{L}f + \int\limits_{-\infty}^{t} \mathsf{Q}(t - \tau) f(\tau) d\tau, \tag{20}$$

Expressing the operator function $\mathsf{Q}(t)$ in terms of its Mellin transform

$$\mathsf{Q}(t) = \frac{1}{2\pi i} \int\limits_{\sigma - i\infty}^{\sigma + i\infty} t^{-s} \bar{\mathsf{Q}}(s) ds, \ \sigma = \Re s,$$

we substitute this expression into the integral term of Equation (20):

$$\int_{-\infty}^{t} \mathsf{Q}(t - t')f(t')dt' = \int_{\sigma - i\infty}^{\sigma + i\infty} \mathsf{W}(s)\, \mathsf{I}_t^{1-s}f(t)ds.$$

Here

$$\mathsf{W}(s) = \frac{\Gamma(1 - s)}{2\pi i}\bar{\mathsf{Q}}(s),$$

and

$$\mathsf{I}_t^{1-s}f(t) = \frac{1}{\Gamma(1 - s)}\int_{-\infty}^{t} \frac{f(\tau)d\tau}{(t - \tau)^s}$$

is an integral of complex order $\mu = 1 - s$. As a result, we arrive at the equation

$$\frac{\partial f(t)}{\partial t} = \mathsf{L}f(t) + \int_{\sigma - i\infty}^{\sigma + i\infty} \mathsf{W}(s)\mathsf{I}_t^{1-s}f(t)ds, \tag{21}$$

containing an operator that can be interpreted as an integral operator with *operator-distributed* complex order. The transition from the integral to fractional derivative can be realized, say, by regularization with respect to Hadamar (by selecting the final part for $\mu < 0$).

Based on the foregoing, the following conclusion can be drawn: in contrast to a closed Hamiltonian system controlled by a differential equation of integer order, its subsystem is controlled by an integro-differential equation of a fractional (distributed along a contour in the complex plane) order. It is the spectral operator-function that determines the specifics of the kinetics of an open system of this class, and, in our opinion, an important direction in the dynamics of open systems lies in the development of the mathematical apparatus necessary for calculating and approximating the spectral function $\mathsf{W}(s)$ [15].

In a special case of a single-point spectrum, Equation (21) can be reduced to renewal integral equation

$$f_\nu(t) = \int_0^t \psi_\nu(t - t')f_\nu(t')dt' + \psi_\nu(t), \tag{22}$$

with the kernel

$$\psi_\nu(t) = \mu t^{\nu - 1}\sum_{j=0}^{\infty} \frac{(-\mu t^\nu)^j}{\Gamma(\nu j + \nu)} = \mu t^{\nu - 1}E_{\nu,\nu}(-\lambda t^\nu), \quad t > 0, \tag{23}$$

posessing the power time-asimptotics (Figure 4). Omitting technical details, we represent here result of these transformation in the form of a system of fractional-order differential equation

$$_0\mathsf{D}_t^\nu p_n(t) = \mu[p_{n-1}(t) - p_n(t)] + \frac{t^{-\nu}}{\Gamma(1 - \nu)}\delta_{n0}, \qquad 0 < \nu < 1.$$

for probability $p_n(t)$, that $n$ aftershocks within time $t$ after the mainshock will occur. This is the fractional Poisson process, When $\nu = 1$, it becomes an ordinary Poisson flow of pulses with independent exponentially distributed interval between them. However, if $\nu < 1$, the kernel grows a long tail, and clusters with gaps appear in the aftershock flow [16,17].

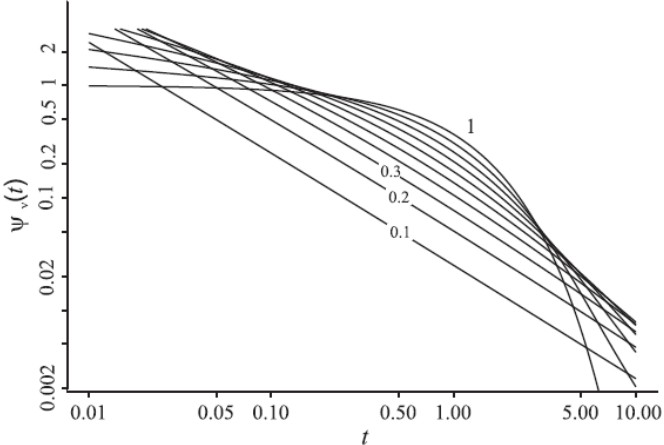

**Figure 4.** Density of distribution of time intervals between shocks.

Using this distribution as a transition probability density for the time-renewal Equation (22) and performing the Laplace transform of the resulting Equation (23), we find in vicinity of $\lambda = 0$:

$$\hat{f}_\nu(\lambda) = \int\limits_0^\infty e^{-\lambda t} f_\nu(t) dt \sim \mu\nu\Gamma(\nu)\lambda^{-\nu}, \quad \lambda \to 0.$$

Finally, applying the Tauberian theorem, we obtain in a long-time asymptotics

$$f_\nu(t) \sim \mu\nu t^{\nu-1}, \quad \langle N_\nu(t) \rangle \sim \mu t^\nu, \quad t \to \infty.$$

### 3.6. Fractional Equations

The random process described by the renewal Equation (23) with transition probability density (22) is called a highlightingfractional Poisson process of order $\nu$ [16,18]. The origin of this term is related to fractional differential nature of the equations describing this process:

$$_0D_t^\nu \psi_\nu(t) + \mu\psi_\nu(t) = \mu\delta(t), \tag{24}$$

where

$$_0D_t^\nu \psi_\nu(t) \equiv \frac{d}{dt} \int\limits_0^t \left\{ \frac{(t-t')^{-\nu}}{\Gamma(1-\nu)} \right\} f(t') dt', \quad 0 < \nu < 1,$$

is the Riemann–Liouville derivative of fractional order $\nu$. Observe, that Equation (24) can be considered as a particular case of a more general Equation (21). relating to the case when its spectrum consists of one isolated point, corresponding to fixed order $\nu$.

### 3.7. Intermittency and Clusterizatioin

In a usual Poisson flow of events, the upscaling (equivalent to an increase in intensity $\mu$) leads to smoothing of fluctuations, the random realization moves closer and closer to a linear function, representing the average number of shocks by the end of the count. Starting from some on a large scale, their difference (that is, the statistical fluctuations of the process) can be neglected. The situation is different in case of fractional-Poisson flow: fractality is self-similarity on different scales, and if we observe a property at one scale, it must be observed at others. Consequently, the fluctuations should not disappear with increasing scale: on any scale there are clusters of pulses interspersed with rarefied, almost empty gaps. This is good, as seen in Figure 5, showing how changing the scale (or keeping the same intensity) by 100 times affects, on the distribution of the number of process events over equal time intervals (bins). In the case of PP (left pair of histograms),

we see a noticeable effect of the scaling procedure, while in the case of fractional Poisson (right pair of histograms) the qualitative difference is practically not visible.

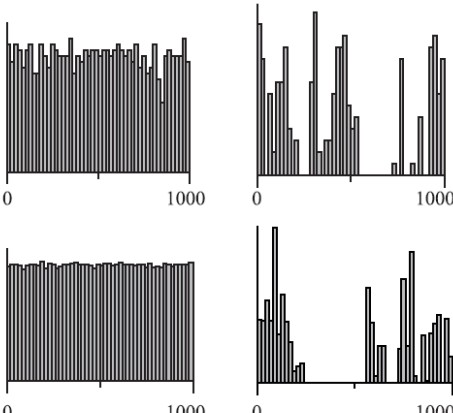

**Figure 5.** A change in the $\mu$ intensity by a factor of 100 noticeably smoothes the Pp histogram (left pair), but has little effect on the fPp histogram (right pair).

Recall that the distribution of the number of events in the Pp over time $t$ obeys the Poisson law:

$$p_n(t) = \frac{\bar{n}^n}{n!}e^{-\bar{n}}, \qquad \bar{n} \equiv \langle N(t) \rangle = \mu t.$$

Introducing the quasi-continuous scaling variable $Z = N/n$, we obtain for its density

$$f(z; \bar{n}) = \bar{n}\frac{\bar{n}^{\bar{n}z}}{\Gamma(\bar{n}z + 1)}e^{-\bar{n}} \sim \sqrt{\frac{\bar{n}}{2\pi}}\exp\left\{-\frac{(z-1)^2}{2/\bar{n}}\right\} \to \delta(z-1), \quad \bar{n} \to \infty.$$

It is clearly seen in Figure 6, as the distribution thicks at point 1 when $\nu$ approches to 1.

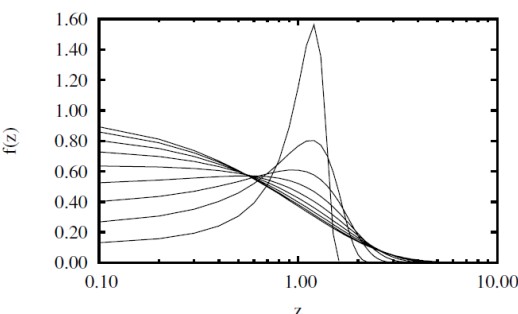

**Figure 6.** The limit $(t \to \infty)$ distribution density of the scaling variable $Z = N/\bar{n}$ for $\nu = 0.1, 0.2, 0.3, 0.4, 0.5, 0.6, 0.7, 0.8, 0.9$ (upwards).

This is just an algebraic translation of the phrase: under the specified condition, the fluctuations are negligible, and the variable $Z$ can be considered as deterministic. It is interesting to note that in the fPp the distribution analogous scaling variable $(\bar{n} = \mu t^\nu)$ does not degenerate as $t \to \infty$ but is expressed in terms of one-sided stable density $g_+^{(\nu)}(x)$ by a simple formula

$$f_\nu(z; \bar{n}) \to f_\nu(z) = \left\{\frac{[\Gamma(\nu+1)]^{1/\nu}}{\nu}g_+^{(\nu)}\left(\left(\frac{\Gamma(\nu+1)}{z}\right)^{1/\nu}\right)z^{-1-1/\nu}\right\},$$

which also admits an elementary representation of the moments

$$\langle Z^k \rangle = [\Gamma(1+\nu)]^k \Gamma(1+k)/\Gamma(1+\nu k)$$

$$\langle Z^k \rangle = \frac{[\Gamma(1+\nu)]^k \Gamma(1+k)}{\Gamma(1+\nu k)}.$$

## 4. Multipower Models of Fractional Responce

### 4.1. One-Power Approximations of Non-Debye Relaxation

The key difference between relaxation processes in disordered inhomogeneous structures lies in their stochastic regime and power laws similar to those that arise in turbulence phenomena. It was shown above that the general nature of the manifestation of such properties may turn out to be not just a consequence of a specific local structure, but the existence of hidden variables that are outside the controlled parameters and have multifractal properties. Such a statistical–mechanical view of the problem makes it easier to accept the fact that those occurring in different mechanically weakly connected regions exhibit qualitatively coinciding properties. From a theoretical point of view, the non-exponential (non-Debye) nature of the relaxations of seismic systems is the causative factor. From a mathematical point of view, this is a consequence of the non-integer (fractional) order of the derivatives describing such relaxation.

Numerous experimental data gathered, for instance, in books [19–21] convincingly show that Debye theory is not able to describe relaxation processes in solids. There exists a few other empirical response functions for solids:

The Cole–Cole (CC) function [22]

$$\tilde{f}_\alpha(i\omega) = \frac{1}{a_0 + a(i\omega)^\alpha}, \qquad 0 < \alpha < 1, \tag{25}$$

the Cole–Davidson (CD) function [20]

$$\tilde{f}^\beta(i\omega) = \frac{1}{[a_0 + a(i\omega)]^\beta}, \qquad 0 < \beta < 1, \tag{26}$$

and the Havriliak–Negami (HN) function [23]

$$\tilde{f}^\beta_\alpha(i\omega) = \frac{1}{[a_0 + a(i\omega)^\alpha]^{\beta/\alpha}}, \quad 0 < \alpha < 1, \quad 0 < \beta < 1 \tag{27}$$

and some others. Below we briefly revew the correspondence between the spectral characteristics of popular relaxation models used in engineering calculations and present the related differential equations with discretely distributed fractional orders. We think that this may be considered as an indication of the existence of real systems to which the generalized (fractional) Liouville theory is applicable.

The HN-function (27) covers the previous cases (25) and (26) and is considered as a general expression for the *universal relaxation law* [20]. A schematic representation of various types of dielectric response in the $\alpha, \beta$ plane is shown in Figure 7. The full vertex of the square represents the Debye relaxation, the full diagonal and the full side display the Cole–Cole relaxation and the Cole–Davidson relaxation respectively.

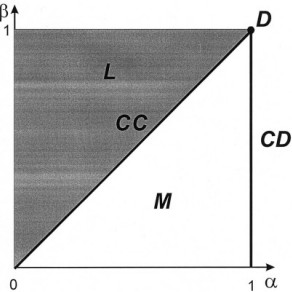

**Figure 7.** Various types of dielectric response. Abbreviations: D—Debye law, CC—Cole–Cole law, CD—Cole–Davidson law, M—region of more typical behavior, L—region of less typical behavior.

The "universality" means that any other relaxation process can be depicted by a point on the $\alpha, \beta$ plane.

*4.2. Three-Power Approximation of HN-Process in Region L*

One can see in [24] what kind of difficulties are to be found in the method of direct transformation of HN-function (27) into a fractional differential equation for the response function $f_\alpha^\beta(t)$. The problem arises because of the nonlinear dependence of the left side of equation

$$[a_0 + a(i\omega)^\alpha]^{\beta/\alpha} \, \tilde{f}_\alpha^\beta(i\omega) = 1$$

on term $(i\omega)^\alpha$ being Fourier-transform of fractional differential operator $D^\alpha$.

However, there exists a simple way to avoid the trouble, at least for region $L$. Indeed, the HN-function is an empirical one and as a result it is an approximate expression for real dependence. Thus, it is not necessary to work with namely this expression: one can find another approximate formula fitting the same experimental data, for example

$$[c_0 + c(i\omega)^\gamma + d(i\omega)^\delta + e(i\omega)^\varepsilon]\tilde{f}_{\gamma\delta\varepsilon}(i\omega) = 1. \tag{28}$$

Numerical calculations show that the HN-function

$$\tilde{f}_\alpha^\beta(z) = \frac{1}{[1 + z^\alpha]^{\beta/\alpha}} \tag{29}$$

can be replaced by the function

$$\tilde{f}_{\gamma\delta\varepsilon}(z) = \frac{1}{c_0 + cz^\gamma + dz^\delta + ez^\varepsilon}, \quad \gamma < \delta < \varepsilon \tag{30}$$

in the case $\alpha < \beta$. The constants $c_0, c, d, e, \gamma, \delta$ and $\varepsilon$ can be determined from the conditions

$$\lim_{z\to\infty} \left[ \tilde{f}_{\gamma\delta\varepsilon}(z) \Big/ \tilde{f}_\alpha^\beta(z) \right] = 1$$

and

$$\lim_{z\to 0} \left[ 1 - \tilde{f}_{\gamma\delta\varepsilon}(z) \right] \Big/ \left[ 1 - \tilde{f}_\alpha^\beta(z) \right] = 1,$$

$$\tilde{f}_{\gamma\delta\varepsilon}(1) = \tilde{f}_\alpha^\beta(1),$$

and

$$\tilde{f}'_{\gamma\delta\varepsilon}(1) = \tilde{f}'^\beta_\alpha(1)$$

The two first conditions yield $c_0 = e = 1$, $c = \beta/\alpha$, and $\gamma = \alpha$. From the third we have

$$d = 2^{\beta/\alpha} - 2 - \beta/\alpha,$$

and from the fourth

$$\delta = \beta \left[ 2^{(\beta-\alpha)} - 2/d \right].$$

The case $\beta = 2\alpha$ is obtained by the limit transition leads to the exact result

$$\tilde{f}_{\gamma\delta\varepsilon}(z) = \frac{1}{(1 + z^\alpha)^2}.$$

Results of comparative calculations for real and imaginary components of function $\tilde{f}(i\omega)$ plotted in Figure 8 show that Equation (30) fit HN-function with an acceptable accuracy.

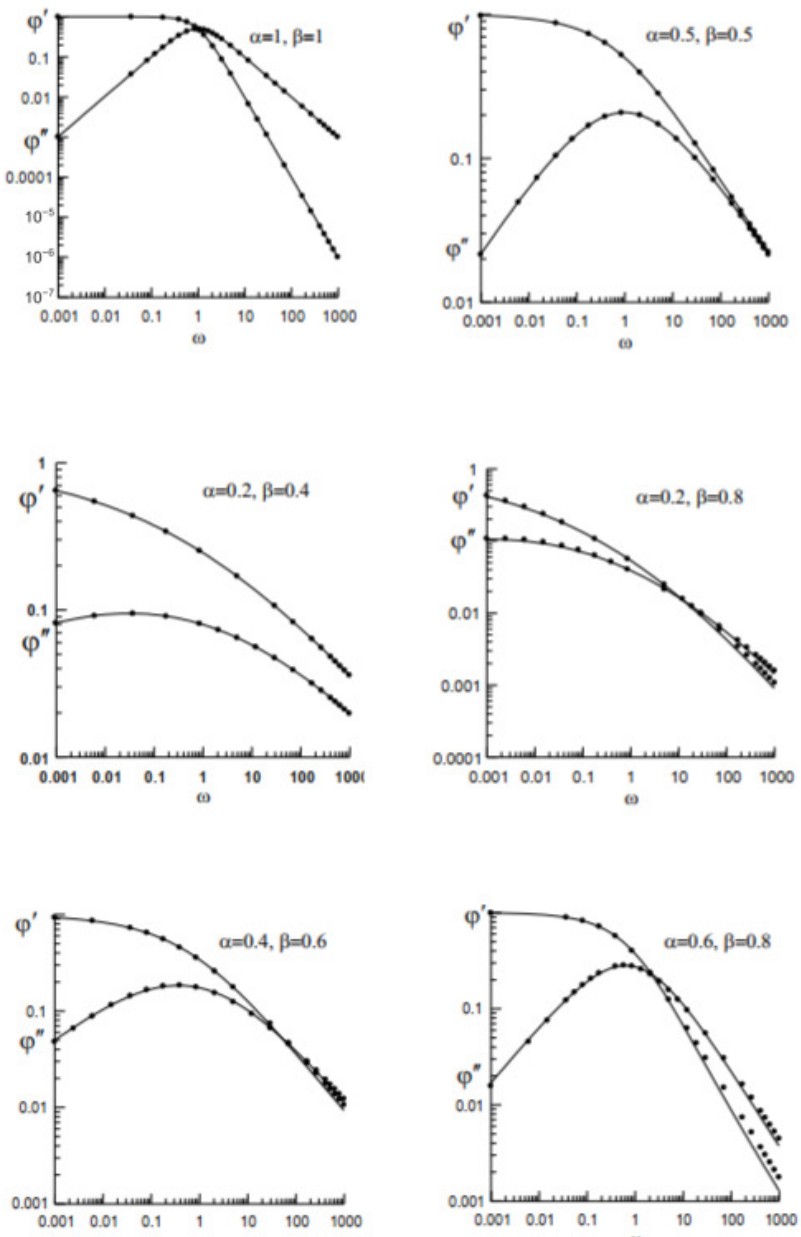

**Figure 8.** Comparison approximation of the Havriliak–Negami law (circles) with exact expression (lines).

Equation (28) represents the differential equation with fractional derivatives

$$\left[c_0 + cD^\gamma + dD^\delta + eD^\varepsilon\right]f_{\gamma\delta\varepsilon}(t) = \delta(t),$$

in Fourier-space. Taking into account $\gamma < \delta < \varepsilon$, we can write $\tilde{f}_{\gamma\delta\varepsilon}(z)$ in the form [25]

$$f_{\gamma\delta\varepsilon}(z) = \frac{1}{ez^\varepsilon + dz^\delta}\frac{1}{1 + \frac{cz^\gamma + c_0}{ez^\varepsilon + dz^\delta}} = \frac{e^{-1}z^{-\delta}}{z^{\varepsilon-\delta} + e^{-1}d}\frac{1}{1 + \frac{e^{-1}cz^{\gamma-\delta} + e^{-1}c_0 z^{-\delta}}{z^{\varepsilon-\delta} + e^{-1}d}} =$$

$$\sum_{m=0}^{\infty}(-1)^m\frac{e^{-1}z^{-\delta}}{(z^{\varepsilon-\delta} + e^{-1}d)^{m+1}}\left(\frac{c}{e}z^{\gamma-\delta} + \frac{c_0}{e}z^{-\delta}\right)^m = \sum_{m=0}^{\infty}(-1)^m\frac{e^{-1}z^{-\delta}}{(z^{\varepsilon-\delta} + e^{-1}d)^{m+1}}$$

$$\sum_{k=0}^{m} \binom{m}{k} \frac{c^k c_0^{m-k}}{e^m} z^{\gamma k - \delta m} = \frac{1}{e} \sum_{m=0}^{\infty} (-1)^m \left(\frac{c_0}{e}\right)^m \sum_{k=0}^{m} \binom{m}{k} \left(\frac{c}{c_0}\right)^k \frac{z^{\gamma k - \delta m - \delta}}{(z^{\varepsilon - \delta} + e^{-1} d)^{m+1}}.$$

The term-by-term inversion based on the general expansion theorem for the Laplace transform yields the final expression for the version of HN function under consideration

$$f_{\gamma \delta t}(t) = \frac{1}{e} \sum_{m=0}^{\infty} \frac{1}{m!} \left(\frac{-c_0}{e}\right)^m \sum_{k=0}^{m} \binom{m}{k} \left(\frac{c}{c_0}\right)^k t^{\varepsilon(m+1) - \gamma k - 1} E_{\varepsilon - \delta, \varepsilon + \delta m - \gamma k}^{(m)} \left(-\frac{d}{e} t^{\varepsilon - \delta}\right).$$

This is a general solution for all relaxation processes belonging to the region $L$.

## 5. Conclusions

In connection with the discussion of the problem of modeling earthquakes (more precisely, aftershocks), we touched here on several fundamental issues—the origin of power laws in geophysics, the concept of self-similarity in the description of disordered systems, the physical content of the concept of heredity, mathematical aspects of the differential calculus of fractional orders and non-Gaussian statistics of limit distributions. Concerning the article [7], devoted to Sol-Gel system description in terms of ladder diagrams, one should stress a mechanical analogy between this system and viscoelastic interacting mantle and crust of the seismological system. Observe that, in both cases, the toy-structures demonstrate large scale self-similarity and inverse-power long-time asymptotics. It is quite reasonable to expect that if one (molecular) model leads to fractional kind of differential equation, then another (seismological) one should reveal the similar hierarchical behaviour, leading to fractional equation as well.

The most promising of the topics covered in the work is the fractional form of the Liouville equation for an open system. Its specific feature is the introduction of a new characteristic—the spectral function of the characteristic exponents. In mathematical terms, this corresponds to the transition to a fractional differential equation of a distributed order. Two non-trivial problems arise in this case: determining the spectral function of a subsystem based on the results of observing its evolution and developing a prediction technology based on spectral analysis. It should be noted that this direction has something in common with the ideas of developing the theory of relaxation based on such equations [26–29].

From the geometric side, the developed approach leads to multifractal analysis, which opens up new prospects for taking into account microstructural inhomogeneities of the medium according to spectral analysis data [30–33].

What is important is that, unlike the equation with a single fixed derivative, this equation describes a monofractal with the corresponding value of the fractal dimension, whereas an equation with a distributed order describes a multifractal medium, characterized by a set (spectrum) of fractal dimensions. The intensity values at different points of the fractal dimension spectrum does not refer to different spatial points of the multifractal (it is homogeneous), but to different scales of its presentation. Of course, this approach may seem to be rather simplified and limited. For example, it does not take into account self-oscillations in nonlinear dynamic systems, which are described by higher-order derivatives, and the number of subsystems in real systems can be very large. The problem of seismicity is essentially related to the problem of turbulence. However, the link is based not only on unpredictable stability in both systems, but also in some sort of self-similar symmetry, which reveals in power type shapes of basic postulates Some aspects of the turbulent diffusion were discussed in the framework of the approach under derivation in [34,35].

**Author Contributions:** Conceptualization, V.U.; methodology, V.U.; software, E.K.; validation, V.U.; formal analysis, V.U. and E.K.; investigation, V.U. and E.K.; data curation, V.U. and E.K.; writing—original draft preparation, V.U. and E.K.; writing—review and editing, V.U. and E.K.; visualization, E.K.; supervision, V.U.; project administration, V.U. All authors have read and agreed to the published version of the manuscript.

**Funding:** This research received no external funding.

**Data Availability Statement:** Not applicable.

**Conflicts of Interest:** The authors declare no conflict of interest. The funders had no role in the design of the study; in the collection, analyses, or interpretation of data; in the writing of the manuscript, or in the decision to publish the results.

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
