# Peer review of "Non-Local Seismo-Dynamics: A Fractional Approach"

_fractalfract, doi:10.3390/fractalfract6090513_

Round 1

Reviewer 1 Report

Review report

The work is devoted to the applicability of fractional calculus for constructing the simplest models of dissipative dynamical systems, which can be used in substantiating the nature of seismicity. Fractional calculus in the description of dynamic systems arises as a consequence of taking into account the specific interaction of subsystems, which has long-range, time-indefinite correlations. Formally, this boils down to replacing the time derivative in the Liouville equation of a dynamical system with a derivative of a fractional order less than one. Physically, this corresponds to the transition from diffusion to subdiffusion in the phase space, which gives a specific relaxation, which the plastic deformations of seismicity have.

Of course, this approach is very simplified and very limited. For example, it does not take into account self-oscillations in nonlinear dynamic systems, which are described by higher-order derivatives, and the number of subsystems in real systems can be very large. The problem of seismicity is essentially related to the problem of turbulence.

Nevertheless, the fractional approach opens up new possibilities, for example, the transition from subdiffusion to superdiffusion with a change in the fractional parameter and an approach to the problem of seismicity from the point of view of the theory of criticality.

This article should be of interest to readers and may be published.

Comments

1. Incorrect reference before formula (3).

2. Line 176, incorrect link to section.

3. In the introduction to the article, the Burridge-Knopov model, very important for understanding seismicity, is not mentioned. This model is about self-oscillations in nonlinear dissipative dynamic systems which are inherently closer to the nature of seismicity than Hamiltonian systems, and a fractional approach can also be applied to them.

4. A composite (multiscale, multistream) Poisson process is used to describe seismicity as a first approximation. The ratios between the streams of random events of various scales are given by the Gutenberg-Richter law. A fractional approach can also be applied to the Poisson process, see https://www.mdpi.com/2504-3110/6/7/372, but dynamical systems are not considered in this case. The use of the fractal approach to dynamical systems may allow one to obtain relationships between the shapes and frequencies of pulsations.

Reviewer

Author Response

«Of course, this approach is very simplified and very limited. For example, it does not take into account self-oscillations in nonlinear dynamic systems, which are described by higher-order derivatives, and the number of subsystems in real systems can be very large. The problem of seismicity is essentially related to the problem of turbulence.»

You are right, the problems are tightly interlinked.  But  the link is based not only on inprediс stability in both systems, but also in some sort of self-similar symmetry, which reveals in power type  shapes of basic  postulates  We insert this remark in Conclusion with link to my article on turbulent diffusion

  1. Incorrect reference before formula (3).

Corrected

  1. Line 176, incorrect link to section.

Corrected

  1. In the introduction to the article, the Burridge-Knopov model, very important for understanding seismicity, is not mentioned. This model is about self-oscillations in nonlinear dissipative dynamic systems which are inherently closer to the nature of seismicity than Hamiltonian systems, and a fractional approach can also be applied to them.

We took this into account and add this article to the Ref. list with a short comment (line 30, cite 2).

  1. A composite (multiscale, multistream) Poisson process is used to describe seismicity as a first approximation. The ratios between the streams of random events of various scales are given by the Gutenberg-Richter law. A fractional approach can also be applied to the Poisson process, see https://www.mdpi.com/2504-3110/6/7/372, but dynamical systems are not considered in this case. The use of the fractal approach to dynamical systems may allow one to obtain relationships between the shapes and frequencies of pulsations.

We indicate this work with a short comment (line 82, cite 7).

The table shows all the corrections made.

Reviewer 2 Report

The article considers the fractional form of the Liouville equation for an open system. This general consideration is used for the statistical estimation of parameters of seismic shocks. All mathematical considerations are presented in an understandable form.

I can recommend the manuscript for publication after some minor corrections:

The title contains the words “seismo-dynamics”. This requires some examples of applying the results obtained to seismological phenomena.

It is strange to see formulas in introduction, especially the formula with parameters that are not defined.

Some typos should be corrected in the revised version. For example,

Line 9 “Prameters”

Line 150 “There are exists”

Line 166 “is” is missing

Line 176 the section number is missing

Line 327 “It became be”

Line 365 “The problem arise”.

After many equations, the paragraph indentation should be removed.

The numbers in the figures are better to make of the same size, for example, Fig. 4 and Fig. 7.

Author Response

"The title contains the words “seismo-dynamics”. This requires some examples of applying the results obtained to seismological phenomena."

We add some discussion of articles directly linked with what reviewr means writing about examples, but we didn't decide overloud the article with special examples, because, as seems to us, the link between power type nature of seismic laws  and  existing non-observable (hidden) parameters derived from the Liouville dynamics potentially covers many special cases.

The bold word motivates the title of the paper, and we don't want to lose it.

Line 9 “Parameters”

prameters  ->   parameters

Line 150 “There are exists”

There are exist

Line 166 “is” is missing

Added  ->   is absent.

Line 176 the section number is missing

added link to section

Line 327 “It became be”

“It became be, and”   replaced  “Consequently, the”

Line 365 “The problem arise”.

Arise -> arises

After many equations, the paragraph indentation should be removed.

according to editorial rules “If the documentclass option "submit" is chosen, please insert a blank line before and after any math environment (equation and eqnarray environments). This ensures correct linenumbering. The blank line should be removed when the documentclass option is changed to "accept" because the text following an equation should not be a new paragraph.”

The numbers in the figures are better to make of the same size, for example, Fig. 4 and Fig. 7.

We have changed the size of the numbers

The table shows all the corrections made.

Reviewer 3 Report

It is important that the authors clearly identify a clear research gap in the paper. In addition,  it may be necessary to validate the theoretical models used in this paper. Please consider improving the work.

Author Response

It is important that the authors clearly identify a clear research gap in the paper. In addition,  it may be necessary to validate the theoretical models used in this paper. Please consider improving the work.

Thank you for the right and useful suggestion. We try to improve it by adding some text.

The table shows all the corrections made.
